# Antiviral Defence Mechanisms during Early Mammalian Development

**DOI:** 10.3390/v16020173

**Published:** 2024-01-24

**Authors:** Felix Mueller, Jeroen Witteveldt, Sara Macias

**Affiliations:** 1Institute of Immunology and Infection Research, School of Biological Sciences, University of Edinburgh, King’s Buildings, Charlotte Auerbach Road, Edinburgh EH9 3FL, UK; f.mueller@sms.ed.ac.uk (F.M.); jeroen.witteveldt@ed.ac.uk (J.W.); 2Centre for Virus Research, MRC-University of Glasgow, Garscube Campus, 464 Bearsden Road, Glasgow G61 1QH, UK

**Keywords:** antiviral, interferon, innate immunity, virus, dsRNA, transposable element, transposon, embryonic stem cells, development, blastocyst

## Abstract

The type-I interferon (IFN) response constitutes the major innate immune pathway against viruses in mammals. Despite its critical importance for antiviral defence, this pathway is inactive during early embryonic development. There seems to be an incompatibility between the IFN response and pluripotency, the ability of embryonic cells to develop into any cell type of an adult organism. Instead, pluripotent cells employ alternative ways to defend against viruses that are typically associated with safeguard mechanisms against transposable elements. The absence of an inducible IFN response in pluripotent cells and the constitutive activation of the alternative antiviral pathways have led to the hypothesis that embryonic cells are highly resistant to viruses. However, some findings challenge this interpretation. We have performed a meta-analysis that suggests that the susceptibility of pluripotent cells to viruses is directly correlated with the presence of receptors or co-receptors for viral adhesion and entry. These results challenge the current view of pluripotent cells as intrinsically resistant to infections and raise the fundamental question of why these cells have sacrificed the major antiviral defence pathway if this renders them susceptible to viruses.

## 1. Introduction

During the early stages of mammalian development, both the cells and embryonic structures critical for supporting embryogenesis and formation of all the tissues of the adult organism are established. After fertilisation, embryonic cells transition from a toti-potent to a pluripotent state during the establishment of the blastocyst [1]. The blastocyst consists of two distinct structures: the inner cell mass from which human and mouse embryonic stem cells (ESCs) are derived and the outer layer of cells, or trophoblast, that will form the foetal component of the placenta. Despite the importance of these early stages in embryonic development, early embryos are unable to utilise one of the most important pathways to defend from viruses, the type-I interferon (IFN) response. In this review, we will focus on the regulation of this innate immune pathway during embryonic development, the use of alternative antiviral strategies by pluripotent cells, and review the critical role of specific RNA-binding proteins in providing antiviral defence during this stage. Our meta-analysis suggests that despite the existence of alternative antiviral pathways, pluripotent cells are not generally resistant to viral infections. We have also found that the susceptibility of pluripotent cells to viruses seems to correlate with the expression of viral receptors and co-receptors. This challenges our current view of pluripotent cells being intrinsically immune to viruses.

## 2. The Type-I IFN Response and Early Development

The major innate immune response against viruses is the type-I IFN response. IFNs are secreted upon infection and are sensed by both the infected and neighbouring cells, resulting in the induction of a large group of interferon-stimulated genes (ISGs) [2]. Besides this immediate cellular response, type I IFNs are also responsible for priming the adaptive immune system through the activation of dendritic cells [3,4]. As triggering the IFN response has profound consequences at both the cellular and organism level, a tight regulation of its induction and consequent downregulation is crucial [5]. To ensure its correct activation, cells have developed specialised sensor-proteins, or pattern recognition receptors (PRRs), capable of recognising different virus-specific hallmarks or pathogen-associated molecular patterns (PAMPs). One of the most prominent viral PAMPs is the virus-derived nucleic acids. PRRs have evolved to recognise types of nucleic acids that are typically not present in mammalian cells or specific cellular compartments and that only accumulate upon viral infection. These include long double-stranded RNAs (dsRNAs), short dsRNAs with either 5′ di- or triphosphate ends and cytoplasmic DNA [6].

Several families of PRRs are involved in the recognition of viral RNA PAMPs. These include the cytoplasmic RIG-I-like receptor family (RLR), composed of the retinoic acid-inducible gene-I (RIG-I), the melanoma differentiation-associated protein 5 (MDA5) and the laboratory of genetics and physiology 2 (LGP2), in addition to the endosomal Toll-like receptor (TLR) family. Aside from these, individual proteins can act as PRRs for viral DNA, such as the cyclic GMP-AMP synthase (cGAS) for cytosolic DNA, the interferon-gamma inducible protein 16 (IFI16) for both nuclear and cytosolic DNA and the absent in melanoma 2 (AIM2) for cytosolic DNA. Upon recognition of the viral nucleic acids, these sensors activate downstream signalling through essential adaptor proteins such as the mitochondrial antiviral signalling (MAVS) protein for the RLR receptors, the TIR domain-containing adaptor molecule 1 (TICAM1) and the MYD88 innate immune signal transduction adaptor (MYD88) for the TLR pathways, and the stimulator of interferon response cGAMP interactor 1 (STING) for cGAS signalling. Although these signalling cascades are different, they all culminate in the expression of type I IFNs through the activation of the interferon regulatory factors 3 and 7 (IRF3/7) [7,8]. After expression and secretion, type-I IFNs signal in an auto- and paracrine manner by binding to the surface type-I IFN receptor (interferon alpha and beta receptor subunit 1 and 2, IFNAR1/2) in both the infected but also neighbouring cells. The binding of IFNs to IFNAR1/2 leads to the phosphorylation and dimerization of STAT1 and 2 (the signal transducer and activator of transcription 1 and 2), which translocate to the nucleus to initiate the transcription of hundreds of ISGs, which are responsible for establishing an antiviral cellular state and stimulating the adaptive immune response (Figure 1A) [9].

Despite the crucial importance of IFNs for antiviral defence, there seems to be an incompatibility between the ability to produce IFNs and pluripotency. For instance, cells derived from murine embryonic teratocarcinomas, which still retain certain pluripotent capacity, cannot produce IFNs in response to viral infections or stimulation with the viral mimic, dsRNA. These cells also appear to have a much-reduced response to stimulation with exogenous IFNs, as pre-treatment with IFNs does not improve protection from viral infection. Interestingly, the ability to produce IFNs and respond to exogenous IFNs is acquired after differentiation [10,11,12]. Similar results were obtained with both mouse and human embryonic stem cells (ESCs) as well as human induced pluripotent stem cells (hiPSCs) [13,14,15,16,17]. The absence of IFN production in hiPSCs is intriguing, as these cells originate from somatic IFN-competent cells but lose the ability to produce IFNs when reprogrammed to become pluripotent. Aside from ESCs, mouse oocytes are also incapable of producing IFNs when stimulated with dsRNAs. Instead, oocytes show an efficient RNA interference (RNAi) response, an antiviral system classically attributed to non-vertebrate organisms, which will be described in more detail in the next section [18,19]. Similar observations have been made in rat spermatogonia and human testis explants, which both fail to induce an IFN response upon viral infection or dsRNA (poly I:C) treatment [20,21,22]. These results suggest that the absence of a functional IFN response is characteristic of pluripotent cells, as well as gametes. Instead, adult stem cells and trophoblasts do appear to have a functional IFN response [23,24]. Aside from viruses and dsRNA, mouse ESCs are also not capable of activating an innate immune response against viral DNA mimics such as G3-YSD, lipopolysaccharide (LPS, which mimics bacterial infections) or bacteria [17,25,26].

The mechanisms responsible for silencing the IFN response in pluripotent cells have still not been completely elucidated. Thus far, low expression of critical PRRs and silencing of key innate immune genes by miRNAs have been associated with a poor ability to produce IFNs by ESCs. For instance, mouse and human ESCs and iPSCs are known to express low levels of MDA5 and TLR3 while maintaining similar levels of RIG-I compared to differentiated cells [13,14,16,27]. In addition, an ESC-specific miRNA was shown to silence MAVS expression in mouse ESCs, leading to inactivation of the whole RLR-sensing pathway. Both removal of the miRNA and ectopic expression of MAVS reconstituted a functional IFN response [17]. A possible mechanism explaining the poor response of hESCs to exogenous IFNs was suggested by Hong and Carmichael [15]. They reported high expression of the suppressor of cytokine signalling 1 (SOCS1) in hESCs, which decreased upon differentiation. Normally, this protein is expressed as part of a negative feedback loop of STAT1 signalling but was found to be responsible for dampening the response to IFNs in hESCs.

## 3. Alternative Antiviral Responses during Early Embryonic Development

The absence of a functional IFN response may pose a strong evolutionary force to develop alternative antiviral mechanisms. Thus far, three distinct alternative antiviral responses have been described in ESCs: antiviral RNAi, ERASE and the constitutive or intrinsic expression of a subset of ISGs (Figure 1B) [28,29,30,31,32]. Due to the high levels of activity of transposable elements (TEs) during early development and the viral origin of some TE families, some of the mechanisms acting to control the potentially deleterious effects of TEs during development also provide some level of antiviral defence [33]. 

### 3.1. Mammalian Antiviral RNAi

Since 2013, several reports have suggested a role for antiviral RNAi in mammalian somatic but also pluripotent cells [28,29,34,35,36]. RNAi is a post-transcriptional gene-silencing mechanism orchestrated by the RNA endonuclease activity of Dicer and the slicing activity of Argonaute (Ago2 in mammals) [37,38,39,40]. Antiviral RNAi starts by Dicer binding and cleaving long viral-derived dsRNAs into 21- to 23-nucleotide long small interfering RNAs (siRNAs). These siRNAs are next loaded onto the RNA-induced silencing complex (RISC) with Ago2, driving recognition by base-pairing of its cognate viral RNA. Upon binding, the viral target RNA is cleaved by Ago2, resulting in reduced viral RNA availability for replication and egress [41,42].

Typically, organisms where RNAi is crucial for antiviral defence express multiple Dicer genes, with forms designated for antiviral RNAi and others specialised in endogenous small RNA production, including microRNAs (miRNAs) [43]. In contrast, mammals express a single Dicer gene (*DICER1*) that is involved in both miRNA and siRNA biogenesis [44,45]. Despite its involvement in both biogenesis pathways, Dicer cleaves precursor miRNA substrates more efficiently than long dsRNAs, from which siRNAs can originate [36]. Interestingly, truncation of the N-terminal helicase domain of human Dicer leads to increased cleavage of perfectly complementary dsRNAs compared to full-length Dicer. Truncated Dicer generates siRNAs that are loaded onto the RISC complex and are able to inhibit the expression of target mRNAs, including viral RNAs [46]. A naturally occurring isoform of Dicer lacking the N-terminal helicase domain (Dicer^O^) was found in mouse oocytes and also exhibited a more efficient cleavage of long dsRNAs compared to full-length Dicer [47]. Another natural, RNAi-proficient isoform of Dicer was detected in both a range of pluripotent and somatic mouse cells, but also in humans, including hESCs, hiPSCs and somatic cell lines [30]. Overexpression of this isoform (aviD) in *DICER1*^−/−^ cells restored miRNA expression but also showed antiviral activity, with decreased Sindbis or Zika virus production. AviD is an alternatively spliced isoform of Dicer, lacking exons 7 and 8, which encode for one of the three helicase domains previously implicated in inhibiting siRNA processivity [46,47,48]. For the efficient processing of miRNA and siRNA substrates, the N-terminal helicase domain of Dicer needs to associate with its endonucleolytic RNase IIIb domain. While the N-terminal helicase domain of non-vertebrates allows for a certain structural flexibility and can change its conformation to accommodate different RNA substrates, the helicase domain of mammalian Dicer retains its conformation during RNA binding, limiting its potential substrates to shorter RNAs, including pre-miRNAs or shRNAs [49,50]. Thus, deletions within the N-terminal domain might help overcome these structural limitations and promote the processing of long dsRNA substrates, including those that are produced during viral replication [30,46,47]. However, whether the RNAi-based antiviral response plays a meaningful role in mammalian antiviral immunity remains unclear, as Dicer inactivation results in both siRNA and miRNA biogenesis inactivation, but also increased IFN responses [17,30,51].

Aside from its antiviral role, RNAi is also involved in the suppression of TEs. This functional overlap possibly stems from the shared evolutionary origin between some TE families and viruses. Knock-down of *Dicer* in early mouse embryos leads to increased levels of two types of autonomous long-terminal repeat (LTR) retrotransposons, the Intracisternal A-Particles (IAP) and the murine endogenous retrovirus-L (MERVL) [52]. This was later corroborated in mESC *Dicer* knockout cell lines and mouse oocytes, which both showed an increased expression of ERVs, but also of other types of TEs, including the long- and short-interspersed nuclear elements (LINEs and SINEs) [47,53,54,55,56,57,58,59]. 

While ESCs can produce IFNs upon differentiation, the relevance of RNAi-mediated antiviral responses is suggested to decrease after differentiation [28,29,30,32,60]. It is, therefore, hypothesised that the IFN and RNAi responses can inhibit each other. Despite this, antiviral RNAi has been detected in non-pluripotent cell lines, including A549, HEK293T, BHK-21 and primary murine lung fibroblasts (MLFs) [34,35]. Supporting the cross-inhibition of RNAi and IFNs, Witteveldt et al. [17] demonstrated that in the absence of the central factor for RNAi, Dicer, mESCs become capable of synthesising IFNs [17]. In turn, inactivation of the IFN response in somatic cells results in detectable RNAi activity [61] and some ISGs, such as LGP2, can inhibit antiviral Dicer activity in somatic cells [62]. The underlying mechanisms behind this apparent incompatibility between the RNAi and IFN response remain to be fully elucidated [7]. Furthermore, the functional role of antiviral RNAi compared to the IFN response remains to be determined. 

### 3.2. Intrinsic Expression of ISGs

ISGs are the ultimate effector proteins that establish the antiviral state in mammalian cells. Most ISGs are generally lowly expressed in homeostatic conditions and are only induced upon activation of the JAK/STAT pathway [63,64]. However, pluripotent cells, including ESCs, adult tissue stem cells and iPSCs, constitutively express a subset of ISGs, suggesting that these cell types display a basal and intrinsic antiviral state [32,60]. This subset of ISGs corresponds to the top 20% of the highest expressed genes in both mouse and human ESCs, and its composition varies in a stage- and species-specific manner. Interestingly, ISGs that are known to have antiproliferative or pro-apoptotic properties, such as *CH25H*, *TNFSF10* or *IFI27*, are not detected in ESCs, as these could interfere with embryogenesis [32,60,65,66,67]. Moreover, activation of the type-I IFN response results in defects with the differentiation of pluripotent cells [16,67]. Twenty-one ISGs have been identified as highly expressed in human ESCs, while 22 ISGs are considered highly expressed in mouse ESCs. Only 10 ISGs are shared between mouse and human ESCs, including members of the IFITM family, which are known to be potent antiviral factors [32,60,68,69,70]. Two of the highly expressed ISGs—*Mov10* in mESCs and *ADAR* in hESCs—are also RNA-binding proteins (RBPs) and are especially interesting, as they are known to be involved in the regulation of both viral- and TE-derived RNAs. These represent additional examples of the functional overlap between TE control and antiviral defence in early development [71,72,73,74,75,76]. 

#### 3.2.1. Adenosine Deaminase RNA Specific (ADAR)

Adar is a deaminase enzyme that acts on dsRNA, converting adenosines (A) to inosines (I) (A-I) [77,78,79,80]. This activity mutates nucleic acid sequences, as well as disrupts dsRNA structures. In human ESCs, ADAR (also known as ADAR1) is one of the highly expressed ISGs in the absence of infection [32]. ADAR can edit both virus- and TE-derived dsRNAs, and depending on the virus, deamination can have pro- or anti-viral consequences [81,82,83,84,85]. ADAR and ADARB1 have been reported to target all types of viruses (RNA and DNA viruses). For instance, Adar has been suggested to act as an antiviral factor against Encephalomyocarditis virus (EMCV) [86]. This virus forms circular RNA intermediates during replication, which are inaccessible for PKR binding and consequent immune responses. Adar can destabilise the stem structure of the circular RNAs by editing specific sequences, so-called reverse complementary matches (RCM), enabling the binding of PKR [87,88].

In addition to viruses, Adar can edit cellular RNAs. For instance, binding and editing of endogenous TE-derived dsRNAs prevents aberrant innate immune activation by MDA5 and PKR. This function is important for embryonic development, as *Adar*^−/−^ mice are embryonically lethal (E11.5-12.5), and the lethality can be partially reverted by knocking out MDA5 and/or MAVS or fully reverted to birth by knocking out both PKR and MDA5. These findings suggest that activation of the IFN response and the mRNA translational shutoff are toxic for embryonic development [89,90,91,92,93,94,95,96,97,98,99,100,101,102,103]. In humans, inactivating mutations of *ADAR* are associated with type-I interferonopathy, Aicardi-Goutières syndrome. Patients show a constitutive activation of the IFN response, probably due to the accumulation of unedited dsRNAs [98,104]. 

Besides its role in immunity, both *ADAR* (*ADAR1*) and *ADARB1* (*ADAR2*) have been shown to restrict human Alu and LINE-1 retrotransposons. Alu elements are primate-specific SINEs that make up to 10% of the human genome and are one of the major cellular targets for Adar [105,106]. Due to their repetitive nature, Alu elements are prone to form dsRNA secondary structures through complementary pairing between copies inserted in sense and antisense orientation. Adar can disrupt these structures through A-to-I conversion, which is reminiscent of the mechanism by which Adar restricts EMCV. LINE-1 is restricted by interactions of Adar(s) with the LINE-1 RNP complex, potentially interfering with the retrotransposition cycle. This function seems to be editing-independent, as both *ADAR* and *ADARB1* mutants lacking deaminase activity remained capable of restricting LINE-1 mobilisation [75,105,106,107,108]. While *ADAR* most likely drives the LINE-1 RNP into stress granules, *ADARB1* acts within the nucleus [107,109]. Although the mechanisms by which Adar(s) inhibit the LINE-1 retrotransposon are not fully clarified, it shows the versatile function of these proteins in viral and TE defence and its importance in restricting aberrant innate immunity in mice and humans [33]. 

#### 3.2.2. Moloney Leukaemia Virus 10 (MOV10)

MOV10 is an ATP-dependent RNA helicase that was originally identified as an antiviral restriction factor against Molony murine leukaemia virus (MMLV) in mice. It is also one of the highly expressed ISGs in mESCs [32,110]. Since its discovery, it has been found to interact with other viruses, including Human Immunodeficiency Virus-1 (HIV-1), Influenza A virus (IAV) and Hepatitis B and C virus (HBV, HCV). The effect of MOV10 during infection appears variable and has been reported to be both pro- and anti-viral, depending on the virus or on the phase of the viral lifecycle that is affected [74]. Besides its role in viral infections, MOV10 has also been reported to strongly repress LINE-1 and IAP retrotransposition by various mechanisms [111,112]. First, MOV10 was shown to impair the retrotranscriptase activity of both IAP and LINE-1 elements [113]. Second, MOV10 in complex with RNase H2 was shown to inhibit LINE-1 retrotransposition by limiting the formation of DNA–RNA heteroduplexes during target-primed reverse transcription (TPRT) [114]. However, these findings contradict the role of RNAse H2 in other settings where it was found to facilitate LINE-1 retrotransposition [115,116]. More recently, MOV10 has been shown to facilitate the addition of non-templated uridines at the 3′-end of LINE-1 RNA by the terminal uridyltransferases (TUTases) 4/7. Introducing uridines at the 3′-end can trigger both LINE-1 RNA decay but also inhibit the TPRT process, as this requires complementarity of the poly-A tail with the T-rich insertion target sites [117]. Importantly, the absence of MOV10 is embryonic lethal, possibly because of its role in restricting potentially deleterious TE activity [112,118]. 

### 3.3. ERASE (Endogenous RTase/RNase H-Mediated Antiviral System)

Recently, an IFN and RNAi-independent antiviral mechanism in mouse ESCs, termed ERASE (Endogenous Reverse transcriptase (RTase)/RNase H-mediated antiviral system), has been identified. This mechanism relies on the ERV-derived reverse transcriptase activity and the endogenous RNase H1 enzyme [31]. RNase H1 belongs to the RNase H superfamily and degrades the RNA strand in RNA–DNA duplexes. Although it is unclear if RNase H1 can also regulate TE activity, it has been shown to partially rescue the LINE-1 retrotransposition defects of RNase H2 knockout cells [115,119]. In ERASE, viral RNAs are first reverse-transcribed to cDNA by the RTases from ERVs, resulting in DNA/RNA hybrids, which are next degraded by the host RNase H1 [31]. This process seems to require specific RTases. While overexpression of the full-length MusD ERV inhibited Encephalomyocarditis virus (EMCV) infection in mESCs, overexpression of the IAPs and LINE-1, which also encode their own RT enzyme, only had a minor effect. Inhibition of MusD-derived RTase by the RTase inhibitor azidothymidine (AZT) diminished the observed antiviral activity. Since mammals encode a wide variety of RTases, Wu et al. hypothesise that other TE-derived RTases might also be capable of acting in this manner [31]. 

## 4. Are Pluripotent Cells Insensitive to Viral Infections?

The presence of ESC-specific antiviral mechanisms and the initially limited number of viruses capable of establishing a successful infection in pluripotent cells led to the general assumption that ESCs are resistant to viruses. In more recent years, an increasing number of successful viral infection models, and even examples of viruses that infect ESCs better than somatic IFN-competent cells, have been reported [17]. An explanation for these observations could be due to one aspect of the viral lifecycle that is rarely addressed, which is the expression of essential (co-) receptors for viral binding and entry on the host cells. There is an enormous variation in the expression of receptors within the various cell types of an organism, and this is one of the main determinants of viral tropism. Here, we have performed a systematic review of the expression of viral receptors in pluripotent cells and compiled a table listing the different viruses used in infection studies in both mouse- and human-derived pluripotent cells (Table 1). For viral receptor expression, human and mouse pluripotent ESC RNA high-throughput sequencing datasets were analysed using DESeq2 [32,120,121]. Log2 normalised counts were used to classify viral receptor expression as follows: (-) not expressed for counts lower than 0.3; (+) lowly expressed for counts between 0.3 and 3; (++) expressed for counts between 3 and 8; (+++) highly expressed for counts over 8; (?) was used if the expression was unknown, or if the gene had no homolog in that particular species. For quantifying infectivity, several methods were considered, including flow cytometry, immunofluorescence and vRNA/vDNA expression data. Infectivity was classified as follows: (-) non-infectious; (+) poorly infectious; (++) infectious and (+++) highly susceptible; (ND) not determined. 

Aside from the observation that there are more viruses able than unable to infect pluripotent cells, this meta-analysis also reveals an additional trend: in cases where the virus cannot or only poorly infects pluripotent cells, at least one of the main respective (co-) receptors is either lowly or not expressed. Conversely, in cases where viruses can establish robust infections in pluripotent cells, their (co-) receptors are all expressed. This latter observation is summarised in Figure 2, where we plotted the infectivity level for each virus on the y-axis against the average receptor expression levels on the x-axis. Despite a number of limitations with this approach, including combining essential and non-essential receptors and the subjective quantification of infectivity, it strongly suggests a correlation between infectivity and receptor expression. Our hypothesis has been experimentally confirmed in only a few reports where both infectivity and receptor expression were quantified in the same pluripotent cell line [126,130,143]. Interestingly, there does not seem to be a relationship between the levels of infectivity and the types of viruses, as dsDNA, ssRNA (+) and ssRNA (-) viruses all vary between infectious and non-infectious types. Viruses from the *Flaviviridae* or *Herpesviridae* family are known for their wide host range [183,184], but infectivity for both families in pluripotent cells is absent or very poor. This lack of infectivity seems to correlate with the absence of some of the major receptors (Figure 2). Based on these data, it seems reasonable to conclude that, for many viruses, the lack of the essential factors for binding and entry in pluripotent cells is a major but not only determining factor for their ability to establish successful infections. It, therefore, seems premature to conclude that pluripotent cells have a general resistance to most viruses solely based on the limited number of non-infectious viruses and the presence of alternative antiviral mechanisms. However, it is also inaccurate to conclude that alternative antiviral mechanisms do not provide a certain level of intrinsic antiviral immunity. Infections of mammalian cells, either somatic or pluripotent, with Nodamura virus (NoV), Encephalomyocarditis virus (EMCV) and Influenza A virus (IAV) revealed the presence of viral suppressors of RNAi, suggesting that viruses have also evolved ways to block antiviral RNAi to ensure a robust infection [28,29,35,41]. Knockout of the highly expressed ISGs *IFITM1-3* in ESCs results in a considerable increase in viral susceptibility [32]. Interestingly, the IFITM proteins’ antiviral mode of action involves the inhibition of viral entry [68].

Taken together, viral susceptibility in early development is determined by a wide range of cellular characteristics, starting with the expression of (co-) receptors, an absence of the IFN response and the activity of alternative antiviral mechanisms.

## Figures and Tables

**Figure 1 viruses-16-00173-f001:**
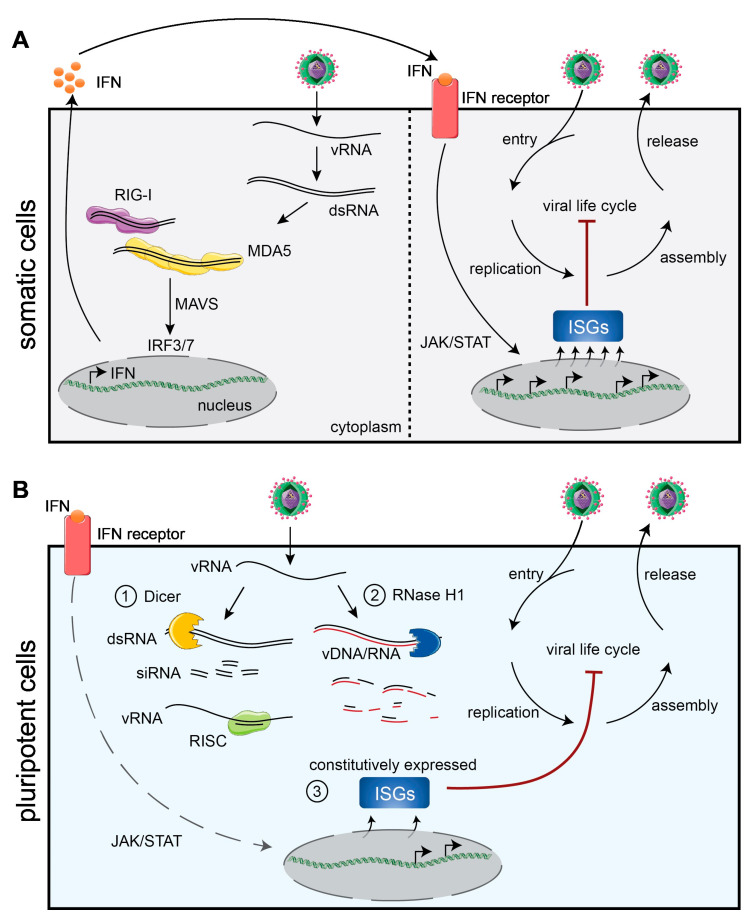
Overview of the antiviral responses of somatic vs. pluripotent cells. (**A**) Somatic cells have specialised cytoplasmic receptors that sense viral double-stranded RNAs (dsRNAs). These sensors include RIG-I and MDA5, which signal through MAVS to activate the transcription factors IRF3/7, leading to the expression of type I IFNs. Secreted IFNs bind the IFNAR1 and 2 receptors, and through JAK/STAT signalling, lead to the expression of hundreds of ISGs and establishment of an antiviral state. (**B**) Contrary to somatic cells, pluripotent cells lack a functional IFN response and fail to produce type I IFNs upon challenge with viruses or viral mimics. Instead, pluripotent cells rely on alternative strategies, including (1) RNA interference, where viral dsRNAs are cleaved by Dicer to generate antiviral siRNAs, which are loaded onto the RNA-induced silencing complex (RISC), (2) RNase H1-mediated degradation of viral RNA (ERASE) after generation of viral DNA/RNA hybrids and, (3) constitutive or intrinsic expression of a subset of ISGs.

**Figure 2 viruses-16-00173-f002:**
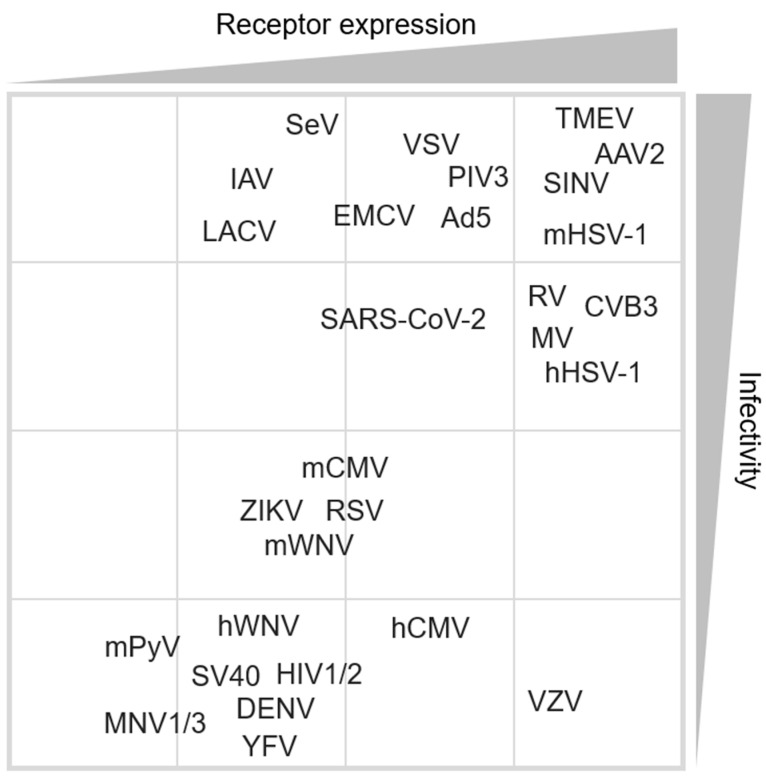
Summary of virus infectivity and (co-) receptor expression in pluripotent human and mouse cells. The figure shows a correlation between susceptibility to viral infections and expression of the viral (co-) receptors. This correlation suggests that the ability of a virus to establish a successful infection is also determined by the ability to enter (grey triangles depict a level of receptor expression or infectivity).

**Table 1 viruses-16-00173-t001:** List of viruses used in pluripotent human and mouse cells, including ESCs, iPSCs and teratocarcinoma-derived cells. The table shows the level of infectivity in the pluripotent human and mouse cells, the (co-) receptors known for each virus and the expression levels (mRNA) of each receptor in human and mouse ESCs.

Virus	Infectivity in Pluripotent Cells		References Infectivity	Receptors	Expression in Pluripotent Cells		References Receptors/Expression
	Human ^1^	Mouse ^1^			Human ^2^	Mouse ^2^	
Adeno-associated virus 2 (AAV2)	+++	+++	[122]	Heparan sulfate proteoglycans	+++	+++	[123,124,125]
				AAVR	+++	+++	
Adenovirus (Ad5)	+++ +++	+++ ND	[122,126]	CAR	+++	+++	[127,128]
				VCAM-1	+	+	
				Heparan sulfate proteoglycans	+++	+++	
				Scavenger receptor A	+++	+++	
Coxsackie B virus (CVB1&3)	++ ++	ND ND	[129,130,131]	CAR	+++	+++	[129,132]
				DAF	+++	+++	
Encephalomyocarditis virus (EMCV)	ND	+++	[29]	Adam9	+++	+++	[133,134,135,136,137]
				Sialic acid (not essential for all strains)	+	+	
				VCAM-1 (only relevant for specific cell types)	+	+	
Herpex simplex virus (HSV-1)	ND ++	+++ ND	[138,139]	Nectin-1	+++	+++	[140]
Human Cytomegalo virus (hCMV)	- - -	- ND ND	[141,142,143]	PDGFRα	++	++	[144,145]
				EGFR	++	++	
				Neurophilin2 (Nrp2)	+++	+++	
				CD46	+++	+	
				BSG (CD147)	+++	+++	
				Thy1 (CD90)	+++	++	
				Heparan sulfate proteoglycans (HSPG)	+++	+++	
				αvβ3 Integrin	++	++	
				CD13	++	+++	
				CD151	+++	+++	
Human immunodeficiency virus (HIV1/2)	-	ND	[146,147]	CD4	++	+	[148,149]
				CCR5	-	+	
				CXCR4	++	+	
Indiana vesiculovirus (VSV)	+++	ND	[32]	LDLR	+++	+++	[150]
				HSP90B1 (Gp96)	+++	+++	
				TLR4	+	+	
				TLR13	?	+	
Influenza A virus (IAV)	ND +++ +++	+++ ND ND	[17,32,151]	Sialic Acid α2,3	+	+	[135,136,152]
La Crosse encephalitis virus (LACV)	ND	+++	[13]	C-type lectins	?	?	[153]
				DC-SIGN (cd209)	+	+	
				Mincle (Clec4E)	+	+	
				Dectin-1 (Clec7a)	+	-	
				Dectin-2 (Clec6a)	+	?	
Measles virus (MV)	++	ND	[131,154]	MSN	+++	+++	[155,156]
				CD46	+++	++	
Mouse Cytomegalo virus (mCMV)	ND ND	+ +	[141,157]	Neurophilin-1 (Nrp1)	++	++	[158]
Mouse Polyoma virus (mPyV)	ND ND	- -	[159,160]	GT1a	?	-	[135,136,161,162]
				GD1a	?	-	
				GT1b	?	+	
				Glycosaminoglycans	+	+	
				ITGA4 (α4β1 integrin)	++	+	
Murine Norovirus (MNV1/3)	ND	-	Personal observation	CD300lf	-	+	[163]
Myxomavirus (MyxV)	-	ND	[164]	Unknown			
Nodamura virus (NoV)	++	ND	[29]	Unknown			
Parainfluenza virus type 3 (PIV3)	+++	ND	[32]	Sialic acid α2-6	+	+	[135,165]
				Nucleolin	+++	+++	
Respiratory syncytial virus (RSV)	+	ND	[32]	Glycosaminoglycans	+	+	[166]
				Nucleolin	+++	+++	
				CX3CR1	+	+	
Rubella virus (RV)	++ ++	ND ND	[131,167]	Sphingomyelin (SGMS1)	+++	+++	[168]
SARS-CoV-2	++	ND	[169]	ACE2	++	+	[170,171]
				TMPRSS2	++	++	
Sendai virus (SeV)	ND	+++	[13]	GYPA	+	+	[172,173]
				ASGR1	++	++	
				Fucosylated glycans	+	+	
				Sialic acid α2-3	+	+	
Simian vacuolating virus 40 (SV40)	ND	-	[159]	Ganglioside GM1	?	-	[135,136,161,162,174,175,176]
				ITGA4 (α4β1 integrin)	++	+	
				Sialic acid	++	++	
				glycosaminoglycans	+	+	
Sindbis virus (SINV)	ND	+++	Personal observation	NRAMP2 (SLC11A2)	+++	+++	[177,178]
				VLDLR	+++	++	
				LRP8 (ApoER2)	+++	+++	
Theiler’s encephalo myelitis virus (TMEV(GDVII)	ND	+++	[17]	Heparan sulfate (essential co-receptor)	+++	+++	[179]
West Nile virus (WNV) Yellow fever virus (YFV) Zika virus (ZIKV) Dengue virus (DENV)	- - + -	+ ND ND ND	[13,32]	αvβ3 Integrin/ITGAV	+++	+++	[180,181,182]
				DC-Sign (CD209)	+	+	
				HAVCR1 (TIM-1)	-	+	
				TIMD4 (TIM-4)	-		
				AXL	++	+++	
				TYRO3	+++	+++	
				Mertk	+	+	
Varicella zoster virus (VZV)	-	ND	[139]	αV integrin/ITGAV	+++	+++	[140]

^1^ Infectivity score for human and mouse pluripotent cells was obtained using data generated by several methods, including flow cytometry, immunofluorescence and vRNA/vDNA expression data. Infectivity is classified as follows: (-) non-infectious; (+) poorly infectious; (++) infectious; (+++) highly susceptible; (ND) not determined. ^2^ Expression levels of the known (co-) receptors in human and mouse pluripotent cells were extracted from RNA-seq data [32,120]. To this end, log2 normalised counts after DESeq2 analysis were used. Receptor level expression is classified as follows: (-) not expressed (log2 norm counts < 0.3); (+) lowly expressed (log2 norm counts 0.3–3); (++) expressed (log2 norm counts 3–8); (+++) highly expressed (log2 norm counts >8); (?) if the expression is unknown, or if the gene has no homologous in that particular species.

## Data Availability

Figures have been designed using ‘Smart Servier Medical Art’ (https://smart.servier.com) under a CC BY 3.0 (https://creativecommons.org/licenses/by/3.0/), accessed on 10 October 2023.

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
