# Peer review of "Antiviral Defence Mechanisms during Early Mammalian Development"

_viruses, 2024, doi:10.3390/v16020173_

Round 1

Reviewer 1 Report

Comments and Suggestions for Authors

This is a very interesting review that should generate debate about the supposed inherent resistance of embryonic and pluripotent cells to viral infection.  The review was well-written and should be published in its present form.  Because of the good quality of the manuscript, I can only suggest a few very minor text corrections.

Line 36: Make "Interferon" lower case.

Line 141: Add "These" before "SiRNAs" and then make "SiRNAs" lower case.

Line 260: Change "virus'" to "viral".

Lines 268 and 269: Either add a space or a hyphen between "3'" and "end".

Line 276: Define "RTase" as "Reverse Transcriptase" somewhere in the sentence, to aid the reader.

Line 283: Remove "'s" from "host's".

Line 296: Change "a virus'" to "the viral".

Line 302: Remove extra space between "." and "The".

Reviewer 2 Report

Comments and Suggestions for Authors

The study by Mueller, Witteveldt and Marcias addresses an interesting and relevant aspect: antiviral defence mechanisms in pluripotent stem cells in the absence of type I interferon-based signalling.

The following points needs to be addressed:

The manuscript provides a very good summary of the generation of interferons and the interferon-based signalling pathway. Figure 1 could be separated into A and B, as they are refenced in the manuscript text as two parts. MAVS is included in the manuscript text and could thus be included in Figure 1. As a general aspect, the authors should briefly highlight the effect of interferons on differentiation capacity of pluripotent stem cells as for example outlined by Eggenberger et al., a publication already included in the refence list. This also relates to line 171 to line 176, which emphasize DICER not only as an antiviral factor, but also as a means to regulate interferon response.

Line 20: The statement on “susceptibility of pluripotent cells to viruses is directly correlated with the presence of receptors or co-receptors for viral adhesion and entry” does not hold true for all viruses (e.g. VEEV as outlined hereafter) and expression of receptors and co-receptors is not the only aspect for a productive infection cycle.

Line 39 to 41: The sentence could also include the intrinsic expression of ISGs in the absence of interferon signalling besides alternative antiviral pathways.

Line 174-176 and 190-191 are repetitive.

Line 224: A reference is missing for the impact of ADAR on EMCV.

Line 295: Is this the correct reference as Witteveldt et al. show that in comparison to wild-type ESC infection with IAV and TMEV was reduced in Dicer-/- ESCs.

The ISGs specifically addressed by the authors were selected based on their nucleic acid-directed antiviral activity?

The authors especially need to revise table 1. The column References for Receptors/expression includes for some viruses the expression of the receptor in stem cells or the reference for the receptor in general. This is rather difficult to follow. For VSV, MV, VEEV reference 32 (Wu et al.) describes a low rate of infectivity or even the absence of infectivity, while in table 1 +++ indicates a high level of susceptibility. VEEV for example would be an exception to the receptor-infectivity correlation (Figure 2). Suitable references for infection of MV on pluripotent stem cells are for example 1: Naaman H, Rabinski T, Yizhak A, Mizrahi S, Avni YS, Taube R, Rager B, Weinstein Y, Rall G, Gopas J, Ofir R. Measles Virus Persistent Infection of Human Induced Pluripotent Stem Cells. Cell Reprogram. 2018 Feb;20(1):17-26. doi: 10.1089/cell.2017.0034. and 2: Hübner D, Jahn K, Pinkert S, Böhnke J, Jung M, Fechner H, Rujescu D, Liebert UG, Claus C. Infection of iPSC Lines with Miscarriage-Associated Coxsackievirus and Measles Virus and Teratogenic Rubella Virus as a Model for Viral Impairment of Early Human Embryogenesis. ACS Infect Dis. 2017 Dec 8;3(12):886-897. doi: 10.1021/acsinfecdis.7b00103.

The table in the manuscript includes the statement "own observations", which is not possible to make at this point.

Reviewer 3 Report

Comments and Suggestions for Authors

In this manuscript, the authors provided an overview in terms of the difference of antiviral mechanisms between somatic and pluripotent cells. In addition, the authors performed meta-analysis to see expression levels of viral receptors in pluripotent cells and found the positive correlation between susceptibility of pluripotent cells to viruses and expression of viral receptors and co-receptors. The manuscript is well written and organized in the first part. In the second part, conclusion is very interesting to me and probably to readers in the journal as well, however, I have no idea about the criteria to define the grade of the infectivity and the expression level. For example, how did you define AXL as a highly expressed one in human and mouse pluripotent cells? And I don’t think that infectivity and expression levels were determined in parallel. In Figure 2, how did you make this graph? How did you compare the infectivity of viruses and define the virus as high or low? Did these infection assay perform with same MOI? There is absolutely NO explanation for Table 2 and Figure 2. Without clarifying all of these, the manuscript should not be published.

Reviewer 4 Report

Comments and Suggestions for Authors

The review explores the significance of the type I interferon (IFN) response as a primary innate immune pathway against viruses in mammals. Despite its crucial role in antiviral defense, this pathway is inactive during early embryonic development. This review underscores the perceived incompatibility between the IFN response and pluripotency in embryonic cells, which possess the ability to develop into any cell type. Pluripotent cells, lacking an inducible IFN response, employ alternative defense mechanisms associated with safeguarding against transposable elements.

While the review is clearly and well-written, there are a few minor comments:

  1. The hyphenation of some words appears unusual at times. For example:
    • Line 30-31: 'totip-otent'
    • Line 168-169: 'pre-miR-NAs'
    • Line 254: 'an-tiviral'
    • Line 329: 'ter-atocarcinoma-derived cells'
  2. The figure legend of Figure 1 would benefit from more detailed explanations. Labelling the RNA as '(v)RNA' and '(v)dsRNA' may imply that host dsRNA in the cytoplasm is normal. However, host dsRNA is not in the cytoplasm unless from viral infection (or poly I:C treatment) and then the antiviral response triggered. Therefore, removing the brackets around the 'v' might help. Additionally, RISC is not described as the RNA-Induced Silencing Complex.
  3. Line 111: It was mentioned that the dsRNA mimic was performed with poly I:C. To maintain consistency, it would be beneficial to specify what was used to mimic viral DNA.

Round 2

Reviewer 3 Report

Comments and Suggestions for Authors

The authors addressed the reviewer's concerns. I have no additional comment.